# A Proteomic Study Suggests Stress Granules as New Potential Actors in Radiation-Induced Bystander Effects

**DOI:** 10.3390/ijms22157957

**Published:** 2021-07-26

**Authors:** Mihaela Tudor, Antoine Gilbert, Charlotte Lepleux, Mihaela Temelie, Sonia Hem, Jean Armengaud, Emilie Brotin, Siamak Haghdoost, Diana Savu, François Chevalier

**Affiliations:** 1Department of Life and Environmental Physics, HoriaHulubei National Institute of Physics and Nuclear Engineering, 077125 Magurele, Romania; mihaela.tudor@nipne.ro (M.T.); mihaela.temelie@nipne.ro (M.T.); dsavu@nipne.ro (D.S.); 2Faculty of Biology, University of Bucharest, 050095 Bucharest, Romania; 3UMR6252 CIMAP, Team Applications in Radiobiology with Accelerated Ions, CEA-CNRS-ENSICAEN-Université de Caen Normandie, 14000 Caen, France; antoine.gilbert@ganil.fr (A.G.); charlotte.lepleux@gmail.com (C.L.); siamak.haghdoost@ganil.fr (S.H.); 4BPMP, Montpellier University, CNRS, INRAE, Institut Agro, 34000 Montpellier, France; sonia.hem@supagro.inra.fr; 5Université Paris-Saclay, CEA, INRAE, Département Médicaments et Technologies pour la Santé (DMTS), SPI, 30200 Bagnols-sur-Cèze, France; jean.armengaud@cea.fr; 6ImpedanCELL Platform, Federative Structure 4206 ICORE, NormandieUniv, UNICAEN, Inserm U1086 ANTICIPE, Biology and Innovative Therapeutics for Ovarian Cancers Group (BioTICLA), Comprehensive Cancer Center F. Baclesse, 14000 Caen, France; e.brotin@baclesse.unicancer.fr

**Keywords:** chondrosarcoma, bystander signaling, proteomic analysis, secretome, stress granules

## Abstract

Besides the direct effects of radiations, indirect effects are observed within the surrounding non-irradiated area; irradiated cells relay stress signals in this close proximity, inducing the so-called radiation-induced bystander effect. These signals received by neighboring unirradiated cells induce specific responses similar with those of direct irradiated cells. To understand the cellular response of bystander cells, we performed a 2D gel-based proteomic study of the chondrocytes receiving the conditioned medium of low-dose irradiated chondrosarcoma cells. The conditioned medium was directly analyzed by mass spectrometry in order to identify candidate bystander factors involved in the signal transmission. The proteomic analysis of the bystander chondrocytes highlighted 20 proteins spots that were significantly modified at low dose, implicating several cellular mechanisms, such as oxidative stress responses, cellular motility, and exosomes pathways. In addition, the secretomic analysis revealed that the abundance of 40 proteins in the conditioned medium of 0.1 Gy irradiated chondrosarcoma cells was significantly modified, as compared with the conditioned medium of non-irradiated cells. A large cluster of proteins involved in stress granules and several proteins involved in the cellular response to DNA damage stimuli were increased in the 0.1 Gy condition. Several of these candidates and cellular mechanisms were confirmed by functional analysis, such as 8-oxodG quantification, western blot, and wound-healing migration tests. Taken together, these results shed new lights on the complexity of the radiation-induced bystander effects and the large variety of the cellular and molecular mechanisms involved, including the identification of a new potential actor, namely the stress granules.

## 1. Introduction

Healthy normal tissues protection and patient recovery without sequelæ are key factors in modern cancer radiation therapy (RT). Certainly, radiation-induced side effects raise some concern due to the subsequent growth in morbidity among paediatric and adult patients. Models used in RT were initially developed from data collected after photon radiation. Emerging protocols of RT with protons or heavier particle, such as carbon ions in advanced medical facilities, have widely changed the way of thinking about local tumor control and the impact on healthy tissues [1,2]. Particle therapy (hadrontherapy) with protons has the advantage of a minimal exit dose after energy deposition in the target volume, and hence better sparing of critical structures in the vicinity of the tumor. Moreover, RT with carbon ions represents an exciting radiation modality, which combines the physical advantages of protons, excepting for an exit fragmentation tail, with higher radiobiological effectiveness [3]. Carbon ion therapy is expected to diminish the radiation morbidity rate. However, the multitude of combinations of radiation quality (linear energy transfer, energy, dose rate, dose, etc.) and tissue biological status (cell culture conditions, genetic background, etc.) does not ease the building of a relevant model for healthy tissue or tumor exposure during RT.

Irradiated cells may release signals which can induce biological alterations of neighboring non-irradiated cells termed the “radiation-induced bystander effects” (RIBE) [4,5]. There is no concord on a precise designation of RIBE, which involves distinct signal-mediated effects within or outside the irradiated volume [6]. Several cellular mechanisms have been suggested to be involved in the transmission of bystander signals by irradiated cells, including the secretion of soluble factors in the extracellular matrix, or the direct communication via gap junctions [7]. This phenomenon was observed in vivo in a context of major local inflammation, linked with a global imbalance of oxidative metabolism that makes its analysis challenging using in vitro model systems [8].

Several studies have aimed to identify the mechanisms of the radiation-induced bystander effect using proteomic tools. The protein composition of exosomes secreted by irradiated UM-SCC6 (human head-and-neck cancer cells) was investigated by direct mass spectrometry analysis [9], and showed a large number of proteins modulated (425 up-regulated and 47 down-regulated), belonging to different cellular processes, including the response to radiation, the metabolism of radical oxygen species and the DNA repair. Several soluble factors secreted from irradiated WEHI 164 (mouse fibrosarcoma cell line) were identified with a proteomic approach [10], including heat shock cognate, annexin A1, angiopoietin-2, and stress-induced phosphoprotein 1. The same proteomic approach was used to study a bystander communication between irradiated and non-irradiated fish [11]. In bystander fish, several modulated proteins were similar to those induced in irradiated fish, including hemoglobin subunit beta and hyperosmotic glycine-rich protein.

In one of our previous studies, we used a medium transfer approach to study RIBE [12]. Chondrosarcoma cells were irradiated with X-rays or C-ions (0.05 to 8 Gy) and then the supernatant, containing the signals emitted by irradiated cells, was transferred into flasks with non-irradiated chondrocytes. We use different technical strategies, such as clonogenic assay, multiplex ELISA analysis of conditioned medium, and flow cytometry for cell cycle analysis of direct irradiated and bystander cells. Our results showed a significant reduction in chondrocyte survival after transfer of the conditioned medium from chondrosarcoma cells irradiated with low doses (0.05 and 0.01 Gy) of X-rays and C-ions. By diluting this medium, the phenomenon decreased proportionally, confirming the presence of bystander factors. Some of these factors were partially observed using multiplex analysis of cell cytokines. Taken together, these results showed the capacity of chondrosarcoma cells to secrete bystander signals, particularly at a low irradiation dose, and the capacity of chondrocyte cells to receive these signals [12].

The goal of this study was to better understand the intercellular communication between the irradiated chondrosarcoma cells and the bystander chondrocytes using proteomics [5]. These approaches allowed us to propose new bystander candidates and cellular responses potentially involved in these non-targeted effects. Some of these results were presented at the 45th Annual Meeting of the European Radiation Research Society in Lund, Sweden [13].

## 2. Results

### 2.1. Secretome Analysis of the Conditioned Medium of Low Doses Irradiated Chondrosarcoma Cells

The dataset acquired on the 12 samples comprises 1,338,540 high-resolution MS/MS spectra. First, as we expected, the presence of contaminants due to fetal calf serum, an interpretation of the MS/MS spectra dataset against the “Bostaurus” theoretical annotated genome was performed, prior to interrogating the “Homo sapiens” theoretical coding sequences.

A total of 357,084 MS/MS spectra were interpreted in this first search round. We identified a large number of bovine serum proteins: 889 were validated with at least two peptides of different sequences. The most abundant proteins were: serum albumin with a 21,209 spectral count (i.e., 6% of the total), but this ratio is by far lower than for a serum analysis which shows that the washes were effective but far to be sufficiently exhaustive to remove all traces of bovine proteins. It is interesting to note the BSA rate for each sample: around 2% for the six samples (3 samples 0 Gy and 3 samples 0.1 Gy) with x10 PBS washes to 10–12% for the other samples with x5 PBS washes, showing the necessity to extensively wash the cells prior to incubation and extraction of the secreted proteins.

In a second search round, we analyzed the yet unassigned MS/MSspectra against the SwissProt human database. In this case, a total of 547,589 MS/MS spectra were unambiguously assigned to peptide sequences. With this dataset, we validated 1,522 additional proteins identified with at least two distinct peptides. So, we observed a greater number of proteins compared to the “Bostaurus” request. This double round interpretation confirmed that we had more than just serum proteins in these samples, and certified the presence of the human proteins.

A comparison was then carried out in order to identify the overabundant proteins in the irradiated comparison (0.1 Gy) versus control condition (three biological replicates x two analytical replicates) following the “PatternLab for proteomics” procedure. Considering the condition with 10X PBS washes, a total of 87 groups of proteins were found significantly modulated (*p*-value ≤ 0.05 and fold change ≥ 1.5×), with a total of 55 more abundant and 32 less abundant proteins. From these protein groups, several accessions associated with bovine origin were removed and, finally, 40 were associated with a known human accession. The abundance of 24 proteins were increased while 16 were decreased in the conditioned medium of irradiated cells (Table 1).

We focused on proteins specifically which significantly increased in the conditioned medium of low-dose irradiated cells. Polyadenylate-binding protein 1 (P11940) was over-secreted 23.8 times in the conditioned medium of SW1353 cells irradiated with 0.1 Gy X-rays when compared with the conditioned medium of non-irradiated SW1353 cells. It is interesting to note that several ribosomal proteins increased in the conditioned medium of low-dose irradiated cells (60S ribosomal protein L34; 60S ribosomal protein L7a; 60S ribosomal protein L8; 40S ribosomal protein S2; 40S ribosomal protein S6; Ubiquitin-40S ribosomal protein S27a).

Several other proteins were identified, in relation with the oxidative response and red/ox status (Acetyl-CoA acetyltransferase; Transmembrane protein 189), cadherin binding (Septin-7), cell migration (Profilin-2) or the response to DNA damage stimulus (E3 ubiquitin-protein ligase RBBP6). Several of these proteins were reported to be involved in extracellular exosomes (glyoxalase domain-containing protein 4; protein HSPD1; S-methyl-5’-thioadenosine phosphorylase; Endoplasmic reticulum aminopeptidase 1).

Forty proteins were statistically modulated in the conditioned medium of chondrosarcoma cells irradiated at a low dose (0.1 Gy), when compared with non-irradiated cells; twenty-four proteins were statistically highly expressed. Some of them were involved in key metabolic pathways and were suspected to participate in radiation-induced bystander signaling. These accessions were analyzed according to potential interaction networks with a STRING functional enrichment analysis (Figure 1). A clear and dense cluster can be observed in the middle of the string network, and the accessions all rely on the ribo-nucleosome compartment (GO:1990904) and the cytoplasmic stress granules (GO:0010494).

### 2.2. Quantification of 8-oxoG in the Conditioned Medium of Low Doses Irradiated Chondrosarcoma Cells

To further study the potential impact of oxidative stress on irradiated cells and their corresponding conditioned medium, a quantification of 8-OXO dG was performed in the conditioned media of SW1353 cells irradiated at different doses (Figure 2). A significant increase in 8-OXO dG concentration was observed in the conditioned media of samples irradiated with 0.1 Gy when compared with non-irradiated samples. The tendency of these 8-OXO dG concentration showed a maximum with 0.1 Gy (about 1.4 ng/mL) and then a decrease with doses of 0.2 and 0.5 Gy to reach the basal level observed with non-irradiated samples (about 0.8 ng/mL).

### 2.3. Whole-Cell Proteome Variations of Chondrocytes in Responses to the Conditioned Medium of Low Doses Irradiated Chondrosarcoma Cells

Quantitative changes in proteins were analyzed by comparing the proteomic map of bystander chondrocytes receiving the conditioned medium of low-dose irradiated chondrosarcoma cells or non-irradiated chondrosarcoma cells (control). A total of 1085 proteins were detected on silver-stained 2D-PAGE gels performed with 250 µg proteins per gel. To analyze the bystander-responsive proteins, significant differences in spot volume (from 25% variation) between control and “treated” samples were assessed and protein spots displaying significant up- or down-expression were regarded as candidates and submitted to MS analysis for identification after trypsin proteolysis.

On the whole, 18 spots, representing 1.6% of all spots on the experiment (Figure 3), showed significant variations (*p* < 0.05); green spots and red spots were over-expressed and under-expressed in the bystander condition, respectively (i.e., cells receiving medium from 0.1 Gy irradiated chondrosarcoma cells). Following a mass spectrometry analysis, 9 proteins were identified as increased (green) and 11 proteins were identified as decreased (red) in the treated sample compared to the control condition (Table 2). Proteins involved in cell-junction and adhesion (Actin, Desmoplakin) as well as cell migration (Microtube-associated protein RP, Tropomyosin alpha-1 chain, CAP-G protein) were identified and differentially modulated. In addition, several proteins participating in protein secretion and an interleukin signaling pathway (cyclophilin A, PSME1, 60S acidic ribosomal protein P0, Hspa9, 26S proteasome regulatory subunit 7) were observed too. It was also interesting to notice the implication of thioredoxin (involved in cell redox homeostasis; 27% increased) and several proteins related to exosome formation (Keratin type II and eukaryotic translation initiation factor 3 subunit I).

Using this list of altered proteins (Table 2), we analyzed the corresponding accessions according to potential interaction networks with a STRING functional enrichment analysis (Figure 4), as previously performed in the case of the secretomic analysis. Again, a dense cluster with several accessions linked together many times (PPIA, TXN, HSPA9, ENO1, HSPA8, RPLP0, CCT3) could be observed. According to this analysis, several accessions associated with interleukin signaling pathways and extracellular exosomes were observed (Table 2).

### 2.4. Quantification and Validation of Proteomic Biomarkers

To validate the results of the 2D-gel analysis, the abundances of several proteins were assayed with specific antibodies using protein extracts from samples already used in 2D-GELs selected on the basis of biological functions and fold changes. An equivalent amount of proteins from each sample were loaded, and loading controls (alpha-tubulin and GAPDH) were used in addition. As shown in Figure 5, the expression levels for cyclophilin A, thioredoxin, alpha-enolase, RPLP0, HSC70, HSPA9, and CCT3 were analyzed and quantified by western blotting. The seven proteins displayed a coherent modulation when compared with the 2D-gel proteomic analysis. As an example, in the case of cyclophilin A, a fold change of +1.62 (+62%) was observed by 2D-gel analysis, and an increase of +37% was observed by western blotting when comparing the bystander chondrocytes receiving the conditioned medium of chondrosarcoma cells irradiated at 0.1 Gy with the non-irradiated chondrosarcoma cells (Appendix A). Two proteins (cyclophilin A and thioredoxin) were observed as increased in the condition “0.1 Gy” by western blotting analysis; and five proteins (alpha-enolase, RPLP0, HSC70, HSPA9 and CCT3) were observed as decreased in the condition “0.1 Gy” by western blotting analysis. These proteins appear as good biomarker candidates involved in the cellular response in bystander cells.

### 2.5. Chondrocyte Motility in Extracellular Matrix Affected by Exogenous Stresses

When taken at 24 h, no difference can be observed on chondrocytes between the conditioned medium of control and 0.1 Gy irradiated chondrosarcoma (Figure 6). On the contrary, when taken at 6 h, a significant difference was observed between the conditioned medium of control and 0.1 Gy irradiated chondrosarcoma. Indeed, chondrocyte motility significantly increased from the time points 5 h to 13 h using the conditioned medium of chondrosarcoma irradiated with 0.1 Gy.

## 3. Discussion

The aim of the present proteomic work was to highlight: (i) potential new effectors in the radiation-induced bystander effect, using a comparative secretomic analysis of conditioned media, and (ii) the corresponding cellular response in bystander cells, using a comparative gel-based proteomic analysis of bystander cells. This double strategy is pertinent for applications required without a priori analysis of different cellular compartments, within the same cellular system [5].

A highly enriched compartment, in relation with stress granules (SGs), was observed following our secretomic analysis. Stress granules are described to be non-membrane bound cytoplasmic entities, and are formed following a cellular stress to minimize damages and promote cell survival. Many membrane-less organelles exist within the cell cytosol such as SGs and P-bodies. These organelle forms are commonly referred to as bio molecular condensates [14]. The formation of SGs has been suggested to regulate gene expression during stress. These assemblies sequester specific proteins and RNAs during stress, thereby providing a layer of post-transcriptional gene adaptation with the potential to affect directly mRNA levels, protein translation, and cell survival [15]. In addition to mRNA, SGs mainly contain 40S ribosomal subunits, translation initiation factors such as eIF4G, and RNA-binding proteins (RBPs) [16]. Different cellular stresses can promote SGs formation, such as endogenous stress (hypoxia, low nutrients) and environmental stressors (genotoxic drugs, heat shock, oxidants, or radiations) [17,18]. Exposure of cells to low doses of UVC induces the formation of SGs [19]; according to this study, cells were blocked in G1 phase of the cell cycle in order to repair DNA damages induced by UVC irradiation, simultaneously to the accumulation of the SGs in the cytoplasm. Such significant enrichment of stress granule proteins in the conditioned medium of chondrosarcoma cells irradiated with 0.1 Gy of X-rays is completely unexpected for several reasons: (i) SGs were never described to participate to any intercellular communication, nor bystander effects, (ii) SGs were not described to be secreted by cells, specifically (as cell cytokines) or non-specifically (in cell cargo, such as exosomes, or after cell death), (iii) SGs were not described to display any biological activity with the capacity to transmit a cellular stress to other cells, until now, these biomolecular condensates were supposed to act as a protective structure of protein translation machinery [14]. For all these reasons, it is mandatory to take this result carefully, and not to give definitive conclusions before any additional experiments performed with other cell lines.

In addition to SGs-related proteins, several other protein groups were observed with a significant enrichment in the conditioned medium of chondrosarcoma cells irradiated with 0.1 Gy. Several proteins already observed in exosomes were identified (glyoxalase domain-containing protein 4, uncharacterized protein HSPD1, omega-amidase NIT2, S-methyl-5’-thioadenosine phosphorylase, and rRNA 2’-O-methyltransferase fibrillarin). Such proteins, without any direct biological links between them, could be involved in exosome traffic, thus reinforcing the potential role of exosomes in radiation-induced bystander effects. The lack of knowledge regarding several proteins such as HSPD1 highlights the difficulty to understand and characterize such a complex multi-parameter effect and calls for new experiments aimed at deciphering their functions.

In addition to these unknown potential bystander effectors, several well-known cellular pathways were also observed. Proteins related to oxidative stress were also observed in the conditioned medium of chondrosarcoma cells irradiated with 0.1 Gy. The mitochondrial acetyl-CoA acetyltransferase was observed amongst the most increased protein abundances (in position two in our list), with a four times fold increase as compared with the conditioned medium of non-irradiated chondrosarcoma cells (Table 1). This enzyme is involved in the acetyl-CoA biosynthetic process (GO:0006085) and exerts a central function in the last step of the mitochondrial beta-oxidation pathway, an aerobic process breaking down fatty acids into acetyl-CoA [20]. Its activity is reversible and it can also catalyze the condensation of two acetyl-CoA molecules into aceto-acetyl-CoA [21].

Moreover, the transmembrane protein 189 increased by 3.33 times as compared with the conditioned medium of non-irradiated chondrosarcoma cells. This accession, also named “Plasmanylethanolamine desaturase”, is involved in plasmalogen biogenesis in the endoplasmic reticulum, and is involved in antioxidative (GO:0055114) and signaling mechanisms [22]. An increase in the Profilin 2 protein level with a 2.88 factor was observed. This protein involved in the structure of the cytoskeleton could act as a negative regulator of epithelial cell migration (GO:0010633), as described previously [23]. The reticulocalbin-3 protein (increased 1.64 times) can induce similar effects on cell motility [24]; this protein chaperone exerts an anti-fibrotic activity by negatively regulating the secretion of type I and type III collagens (GO:0032964).

The E3 ubiquitin-protein ligase RBBP6, a protein related to DNA damages, was also observed in the conditioned medium of chondrosarcoma cells irradiated with 0.1 Gy (GO:0006974). This protein, known as being possibly involved in assembly of the p53/TP53-MDM2 complex, results in an increase in MDM2-mediated ubiquitination and degradation of p53/TP53 [25,26], perhaps leading to both apoptosis and cell growth (by similarity) playing a role in the transmission of the radiation-induced bystander effect.

We already observed a bystander cellular response in chondrocyte receiving the conditioned medium of chondrosarcoma cells irradiated with 0.1 Gy of X-rays [12], with a reduction in cell survival and an induction of micronuclei. In order to gain insights into the cell mechanisms and the pathways involved in this bystander response, we analyzed the proteome of the cells using a gel-based proteomic strategy.

While shotgun proteomics may give access to a large list of proteins, 2D-gel-based proteomics allows for the identification of matured proteins, such as the proteolytic cleavage of polypeptide chains or post-translational modifications. Thus, this approach is valuable to assess stress-induced modifications, as already demonstrated by several authors [10,11].

Our findings include the accession identified as cyclophilin A (PPIA = P62937), which is described to be positively regulated with protein secretion and involved in the interleukin 12 (IL-12) signaling pathway. We previously observed this accession as increased in the conditioned medium from irradiated breast cancer cells [27]. IL-12 was defined as a cytokine post-translationally regulated and potentially implicated in the radio-induced apoptotic response in mammary tumor cells. IL-12, observed as increased in bystander cells, could be involved in the propagation of the bystander effect throughout non-irradiated cells.

The accession identified as thioredoxin (TXN = P10599), which is described to be involved in the cell redox homeostasis, was found to have increased in bystander cells according to both the proteomic results and the western blot validation. This factor is believed to contribute to the regulation of transcription factors mediating cellular responses to environmental stress, including radiation [28]. In addition, we observed an increase in 8-oxo-dG in the conditioned medium of low-dose irradiated cells, a nucleotide released by the cells when it is oxidized, which is proof of the presence of oxidative stress. These two mechanisms could be linked in a global oxidative stress response, transmitted to bystander cells, and involved in the cellular response to the bystander effect.

Moreover, besides the identified oxidative stress response, a potential change of cellular motility and migration was observed using a wound healing test on non-irradiated cells receiving the conditioned media of irradiated cells. The bystander cells displayed an increased motility (Figure 5), which could be linked with a decrease in alpha-enolase expression. The nuclear form of the protein was previously identified as Myc-binding protein-1 (MBP1); this form plays a role in the negative regulation of cell growth [29]. Consequently, a decrease in MBP1 could induce an increase in cell motility and migration, at least transitively, as observed in this study after 6 h.

Finally, although SGs were significantly observed as enriched according to our secretomic analysis, no SG-related proteins were observed as modulated in the proteome of bystander cells. One explanation could be that both strategies do not analyze the same cell compartment, i.e., the conditioned medium with the secretomic analysis and the cellular proteome with the proteomic analysis. A second explanation is linked with the biochemical capacities of both strategies: with the secretomic analysis, a gel-free mass spectrometry analysis is performed, allowing for low-abundance and hydrophobic proteins/peptides; on the other hand, with the proteomic analysis, only abundant and soluble proteins can be observed. If SGs are involved in the bystander effects, they can be secreted in the conditioned medium, as observed with our secretomic analysis, but maybe a low amount is able to induce a bystander effect on the non-irradiated cells, which cannot be visualized with our proteomic study.

## 4. Conclusions

Overall, the proteomic analysis underlines the modulation of the abundance of several bystander-related proteins; modulation that was confirmed by western blot and their physiological effects revealed by functional techniques in some instances. The stress granules related to oxidative stress-coping mechanisms were identified for the first time as potential attractive biomarkers of RIBE in the conditioned medium of irradiated cells. The next step of this analysis would be a deep analysis of the role of stress granules in bystander effect transmission, using, for example, cellular models with defects in the stress granules formation processes. In addition, several proteins involved in intercellular signaling, oxidative stress response, and cell motility were also determined in bystander cells in response to such conditioned medium. The results obtained strengthen our previous results concerning the factors involved in the radiation-induced bystander effect at low doses of irradiation, including interleukin.

Taken together, the findings of this study pinpointed the complexity of the mechanisms involved in the radiation-induced bystander effect and the power of a proteomic analysis to bring into light new biomarker candidates of this phenomenon [5].

## 5. Methods

### 5.1. Cell Culture

Two cell lines were used during this study, a chondrosarcoma cell line, SW1353 (CLS Cell Lines Service GmbH, Eppelheim, Germany) and a chondrocyte cell line, T/C28-A2 (gift from Prof. Mary B. Goldring, Hospital for Special Surgery, Weill Medical College of Cornell University, New York, NY, USA), as previously described [12]. These cells were cultured in the same culture medium, minimum essential medium Eagle (MEM, M5650, Sigma-Aldrich, Saint-Louis, MI, USA), supplemented with 5% fetal calf serum, 2 mM L-glutamine, and 1% antibiotics (penicillin–streptomycin solution, Sigma-Aldrich). All experiments were performed in humidified atmosphere with 5% CO_2_ and physioxia conditions with 2% O_2_ at 37 °C, in a Heracell™ 150i Tri-Gas incubator.

The bystander factors, secreted by chondrosarcoma cells, were first evaluated by direct mass spectrometry analysis. The intracellular bystander response in chondrocytes was analyzed using a gel-based strategy.

### 5.2. Experimental Strategy to Characterize the Bystander Effectinduced by Low Doses Irradiated Chondrosarcoma Cells

In order to study the bystander effect between irradiated chondrosarcoma cells (SW1353 cell line) and non-irradiated chondrocytes (T/C28-A2 cell line), we selected a medium-transfer protocol, and we kept the same treatment strategy with all our endpoints (Figure 7). This simplified process allowed us to compare the cell responses of non-irradiated cells, receiving the conditioned medium from irradiated cells.

X-rays irradiations were performed, as previously described [12,30], at room temperature (20 °C) with a tube tension of 225 kV, a copper filter, and an intensity of 1 mA corresponding to a dose rate of 0.2 Gy/min on the Pxi XradSmart 225cX irradiator, dedicated to preclinical research. The dose rate was measured inside flasks with thermoluminescent dosimeters in the irradiation conditions. Thermoluminescent dosimeters were preliminary calibrated on a 15-cm-thick virtual water phantom thanks to reference dose measurements performed with a calibrated ionization chamber following the “American Association of Physicists in Medicine protocol,” developed by the Radiation Therapy Committee Task Group 61, for reference dosimetry of low- and medium-energy X-rays for radiotherapy and radiobiology. The dose rate was finally corrected from the used tube current.

Immediately after irradiation with 0.1 Gy of X-rays, chondrosarcoma cells were cultured with fresh medium. After the incubation period, the conditioned medium was removed from the cells and centrifuged to discard detached cells and cell’s debris.

Then, this conditioned medium was directly analyzed (using three independent biological replicates), or used on non-irradiated chondrocytes to study a potential bystander effect (Figure 7).

### 5.3. Preparation of Conditioned Medium and Shotgun Proteomics Analysis

In the case of the secretome analysis, just before irradiation with X-rays, the complete medium was removed from the flasks and the cell monolayer was extensively washed with PBS. This step was mandatory to reduce the presence in the conditioned medium of bovine serum albumin that could prevent the identification of other proteins. Then, chondrosarcoma cells were irradiated at a low dose (0.1 Gy X-ray) with a serum-free medium. After irradiation, the medium was changed with fresh/serum-free medium and two wash procedures were performed for each irradiation condition and with three independent biological replicates (12 samples). After 24 h, the conditioned medium was removed from the flasks and analyzed by tandem mass spectrometry with technical replicates, thus 24 nanoLC-MS/MS analytical runs.

SW1353 cells were irradiated at confluence and, immediately after irradiation, the monolayer was washed with PBS several times (5× and 10×), and then 3 mL of serum-free medium was added. After 24 h, the conditioned medium was removed from the flasks, centrifuged (2000× *g*), and stored at −80 °C. These experiments were performed in triplicates. Proteins from the 12 samples (2 irradiations, 0 and 0.1 Gy, 2 washing conditions, i.e., X5 and X10 times with PBS, 3 biological replicates) were first precipitated with TCA. For this, 250 µL of trichloroacetic acid at 50% (*w*/*v*) were added to 1 mL of conditioned medium. Precipitated proteins were collected by centrifugation for 15 min at 16,000 g and then dissolved into 30 µL of LDS1X (Invitrogen). The samples were heated at 99 °C for 5 min, briefly centrifuged, and then loaded onto a 4–12% gradient 10-well NuPAGE (Invitrogen) polyacrylamide gel. After a short electrophoresis (5 min), the gel was stained with Coomassie blue safe staining (Invitrogen, Waltham, MA, USA) for 5 min. The polyacrylamide bands corresponding to the whole exoproteomes were sliced and treated with dithiothreitol and iodoacetamide, as recommended [31]. Then, the proteins were subjected to trypsin proteolysis to generate peptides. Each peptide fraction was analyzed twice by nanoLC-MS/MS (analytical duplicates) in data-dependent mode with a Q-Exactive HF (Thermo) mass spectrometer coupled with an Ultimate 3000 chromatography system (Thermo), resulting in 24 runs of high-resolution tandem mass spectrometry. For each peptide fraction, a volume of 10 µL (out of 50 µL) was injected on a nanoscale 500-mm C18 PepMap TM 100 (5 mm × 300 µm I.D., Thermo) column operated, as previously described [32], except that the gradient of acetonitrile (from 4% to 40% of a solution of 80% CH3CN, 20% H20, 0.1% formic acid) was extended to 120 min for deepening the analysis. MS spectra of peptide ions were acquired at a resolution of 60,000. Only peptide ions with 2+ or 3+ charge were selected for fragmentation according to a Top20 method and using a dynamic exclusion of 10 sec. MS/MS spectra of fragment ions were acquired at a resolution of 15,000.

MS/MS spectra were interpreted using the MASCOT search engine, version 2.5.1 (Matrix Science, Boston, MA, USA), with fixed carbamidomethyl modification of cysteines, variable oxidation of methionines, and deamidation of asparagines and glutamines, a maximum of two missed cleavages, mass tolerance of 5 ppm and 0.02 Da on parent ions and secondary ions, respectively. Peptides with a score above the query identity threshold (*p* value below 0.05) were selected and parsed with the Irma software [33]. Only proteins with at least two different peptides were validated. The decoy search option of Mascot was systematically activated to estimate the FDR (<1%). Abundances of the proteins were evaluated based on their spectral counts.

### 5.4. Determination of Extracellular 8-oxo-dG in the Conditioned Media

The media were thawed and 1 mL of each sample was used for the determination of 8-oxo-dG using an ELISA-based method (Health Biomarkers, Stockholm, Sweden, AB). Briefly, one ml of cell culture medium was loaded on a solid-phase-extraction column, followed by a washing step and elution of 8-oxo-dG according to protocols provided by the company Health Biomarker Sweden AB, as previously described [34]. The eluates were concentrated by freeze-drying and dissolved in PBS, pH 7.4, to a volume of 1 mL and the clean-up process was repeated once more to purify 8-oxo-dG. Then, the samples were dissolved in PBS, pH 7.4, to a volume of 1 mL. Based on protocol from kit-provider, 90 μL aliquots of samples were mixed with 50 μL of the primary antibody and transferred to 96-well ELISA plates coated with 8-oxo-dG. After overnight incubation at 4 °C, the plates were washed 3 times by washing solution. Next, 140 μL of HRP-conjugated secondary antibody (goat anti-mouse IgG-HRP, Scandinavian Diagnostic Services, Uppsala, Sweden) was added to each well and incubated for 2 h at room temperature. The wells were washed 3 times with the washing solution. Then, 140 μL of tetramethylbenzidine liquid substrate (ICN BiomedicalsInc, Costa Mesa, CA, USA) was added to each well. The samples were incubated for 15 min at room temperature. The reaction was terminated by adding 70 μL of 2 M H_3_PO_4_ (Merck Millipore, Darmstadt, Germany). The absorbance was read at 450 nm using an automatic ELISA plate reader. All samples were analyzed in triplicate. Standard curves for 8-oxo-dG (from 0.05 up to 10 ng/mL) were established for each plate and the quantity of 8-oxo-dG calculated based on the standard curve and expressed as ng/mL medium.

### 5.5. Medium-Transfer Protocol from Irradiated Cells to Non-Irradiated Cells

Irradiated SW1353 cells and T/C-28a2 bystander cells were plated in T25 cm^2^ flasks at confluence. As previously described [12], immediately after irradiation with X-rays, the medium of irradiated flasks was changed with fresh medium and, after 24 h in contact with irradiated SW1353 cells (to allow the bystander factors to be released), this medium was collected (Figure 7). The condition medium was then centrifuged (2000 g) and transferred in flasks of the same size (T25 cm^2^) containing bystander T/C-28a2 cells at confluence. Bystander cells were kept in contact with the conditioned medium for 24 h and then harvested. The cell pellet was washed with PBS and the dry pellet was kept at 80 °C, until protein extraction.

### 5.6. Gel-Based Proteomic Study of Bystander Chondrocytes

#### 5.6.1. Chemicals

TRIS base, urea, thiourea, CHAPS, iodoacetamide, TEMED, low-melt agarose, Triton X-100, spermine, phosphatase inhibitor cocktail, and bromophenol blue were obtained from Sigma-Aldrich (St. Louis, MO, USA). The protease inhibitor cocktail (Complete Mini EDTA-free) was from Roche Diagnostics (Mannheim, Germany); IPG buffers, IPG strips (pH 4-7) were purchased from VWR (acrylamide was obtained from Bio-Rad (Hercules, CA, USA); and SDS, glycerol, DTT, and TGS 10X were from Eudomedex (Mundolshein, France). All other reagents were of analytical grade.

#### 5.6.2. Protein Extraction and Solubilisation

Proteins were extracted from TC-28/Ac cells (dry pellet) in a sample buffer containing 7 M urea, 2 M thiourea, 4% CHAPS, 0.05% Triton X100, 65 mM DTT, 40 mM spermine, protease, and phosphatase inhibitor cocktails. This suspension was centrifuged at 28,000 g for 60 min, supernatants were collected, and the protein content was estimated using the Bradford method [35]. Proteins were then precipitated using the 2D clean-up kit (GE Healthcare, Chicago, IL, USA) and the pellet was solubilized with TUC solution (7M urea, 2M thiourea, 4% CHAPS) and quantified with the 2D quant kit (GE Healthcare).

#### 5.6.3. Strip Rehydration with Protein Samples: “Sample In-Gel Rehydration”

A protein sample (250 µg) was mixed with rehydration buffer (RB): 7M urea, 2M thiourea, 4% CHAPS, 0.05% triton X100, 0.5% ampholytes (IPG buffer 4–7 GE) and adjusted to the correct volume to rehydrate 18 cm strip (here, 320 µL). Strips were then placed acrylamide face down in the focusing tray equipped with platinum electrode embedded into the running tray (Protean IEF, Bio-Rad, Hercules, CA, USA) and passively re hydrated at 20°C without electricity for 16 h, and then actively rehydrated at 50 V during 9 h, as previously described [27,35,36]. During protein focalization, small electrode wicks were placed between acrylamide and electrode. These paper wicks (Ref 1654071, Electrode wicks, Bio-Rad, Hercules, CA, USA) were, in advance, soaked with water in order to absorb salts and other contaminant species during active rehydration. The IPG strips were then focused according to the following program: 500 V for 1 h, a linear ramp to 1000 V for 1 h, a linear ramp to 10000 V for 33 KV-1 h, and finally 10000 V for 24 KV-1 h.

#### 5.6.4. IPG Strips Equilibration and Second Dimension

The strips were incubated in the first equilibration solution (50 mM Tris–HCl pH 8.8, 6 M urea, 30% (*v*/*v*) glycerol, 2% (*w*/*v*) SDS) with 130 mM DTT, and then in the second equilibration solution (50 mM Tris-HCl pH 8.8, 6 M urea, 30% (*v*/*v*) glycerol, 2% (*w*/*v*) SDS) with 130 mM iodoacetamide.

Strips were then embedded using 1% (*w*/*v*) low-melt agarose on the top of the acrylamide gel and trapped using plastic blockers, as described previously [35]. SDS-PAGE was carried out on a 12% acrylamide gel, using the Dodeca Cell electrophoresis unit (Bio-Rad, Hercules, CA, USA).

#### 5.6.5. Gel Staining and Picture Acquisition

Gels were stained with silver nitrate, as previously described, with some modifications. Briefly, gels were first fixed at least 1 h with 30% ethanol and 5% acetic acid; washed 3 times 10 min with water; sensibilized 1 min with 0.02% sodium thiosulfate; washed 2 min with water; stained 30 min with 0.2% silver nitrate and 0.011% formaldehyde; washed 10 s with water; developed 5 min with 85 mM sodium carbonate, 0.00125% sodium thiosulfate and 0.011% formaldehyde; stopped with 0.33 M TRIS and 1.7% acetic acid; and stored with 5% acetic acid with 2% DMSO [37].

Gels were scanned to images right after staining to limit the polychromatic color of spots. Images were acquired with a GS 800 densitometer (Bio-Rad, Hercules, CA, USA).

#### 5.6.6. Image Analysis

Images from stained gels were analyzed using the Samespots software v4.5 (Non-linear Dynamics, UK). Gels were grouped to create a global analysis with all conditions. Spots of each samples were compared between conditions, and spots were numbered with the same detection parameters, as previously described [35]. A multivariate statistical analysis was performed using the statistic mode of the Samespots software (Non-linear Dynamics, UK). Spots with significant differences (modulation of +/−20% and ANOVA *t*-test *p* < 0.05) were chosen. Spots of interest were selected for subsequent protein identification by mass spectrometry analysis and were picked up using the corresponding preparative silver stained gels.

#### 5.6.7. Mass Spectrometry Analysis of 2D-Spots

Gel spots 2D were manually cut and prepared, as previously described [38]. The LC-MS/MS experiments were performed using a U3000 NCS nano-high-performance liquid chromatography (Thermo Fisher Scientific Inc, Waltham, MA, USA) system and a Q-Exactive Plus Orbitrap mass spectrometer. Next, 6 µL of peptides were loaded onto a pre-column (Thermo Scientific PepMap 100 C18, 5 µm particle size, 100 Å pore size, 300 µm i.d. × 5 mm length) from the Ultimate 3000 autosampler with 0.05% TFA for 3 min at a flow rate of 10 µL/min. Separation of peptides was performed by reverse-phase chromatography at a flow rate of 300 nL/min on a Thermo Scientific reverse-phase nano column (Thermo Scientific PepMap C18, 2 µm particle size, 100 Å pore size, 75 µm i.d. x 50 cm length). After the 3 min period, the column valve was switched to allow elution of peptides from the pre-column onto the analytical column. Solvent A was water + 0.1% FA and solvent B was 80% ACN, 20% water + 0.1% FA. The linear gradient employed was 4–40% of solvent B in 19 min, then 40–90% of solvent B from 19 to 20 min. The total run time was 35 min including a high organic wash step and re-equilibration step. The nanoHPLC and the spectrometer were coupled by nano-electrospray source. Peptides were transferred to the gaseous phase with positive ion electrospray ionization at 1.7 kV. Mass spectrometry data was processed using the Proteome Discoverer software (Version 1.4.0.288, Thermo Fisher Scientific, Bremen, Germany) and the search engine employed in local was Mascot (version.2.4, Matrix Science, Boston, MA, USA). The mass spectrometry data was searched against SwissProt with taxonomy Homo sapiens (20215) with the following parameters: trypsin as enzyme, 1 missed cleavage allowed, and carbamidomethylation of Cystein were used as fixed modifications, and N-terminal acetylation, deamidation of asparagine and glutamine, Nterminal-pyroglutamylation of glutamine and glutamate, oxidation of methionine were used as variable modifications. Mass tolerance was set to 10 ppm on full scans and 0.02 Da for fragment ions. Proteins were validated once they contained at least two peptides with a *p*-value < 0.05) and a false discovery rate <1%.

### 5.7. Western Blotting Analysis

Chondrosarcoma cells (SW1353) were irradiated with a 0.1 Gy dose using the XSTRAHL XRC 160 machine of IFIN-HH, followed by a media transfer to the chondrocyte (T/C28a2) after 24 h, as previously described [12]. At 24 h, after the media transfer, the bystander cells (T/C-28a2) were washed with PBS. The cell pellet was resuspended with homemade RIPA lysis buffer supplemented with protease inhibitors (Roche) and incubated for 30 min on a cold rack, followed by a 15 min centrifugation at 12,000× *g* at room temperature. Protein concentration was determined for all samples with a Bradford assay (Thermo Scientific). Laemmli buffer was added to the sample and denatured at 95 °C for 5 min, followed by a 1 min centrifugation at 16,000 xg. Samples were separated on SDS-poly-acrylamide (15%) gel electrophoresis (SDS-PAGE) using a TV100 electrophoresis unit, run at 110 V for about 1 h, followed by transfer on a PVDF membrane using a TV 100 Electroblotter, at 210 mA for 1 h and blocked with Tris-buffered saline with 0.05% Tween 20 (TBS-T) buffer with 5% milk on slow agitation for 1 h. Membranes were incubated overnight at 4 °C on agitation with the following antibodies: anti-HSPA9 (MA1-91639, Thermo Scientific), anti-HSC70 (PA5-24624, Thermo Scientific), anti-CCT3 (PA5-78953, Thermo Scientific), anti-EN01 (MA5-17627, Thermo Scientific), anti-RPLP0 (PA5-89335, Thermo Scientific), anti-cyclophilin A (39-1100, Thermo Scientific), anti-thioredoxin 1 (MA5-14941, Thermo Scientific), anti α-Tubulin (T5168, Sigma-Aldrich) or GAPDH (sc-32233, Santa Cruz Biotechnology), in the concentrations recommended by the manufacturer. Membranes were than washed 3 times with TBS-T for 10 min, followed by an incubation with specific secondary antibody conjugated with horseradish peroxidase for 2 h at room temperature on agitation, covered from light in concentrations of 1:500 (Goat anti-Rabbit IgG (H+L), 32460, Thermo Scientific, Waltham, MA, USA) and 1:1000 (Goat anti-Mouse IgG (H+L) Poly-HRP, 32230, Thermo Scientific). For α-Tubulin, the primary antibody was diluted 1:30,000 and the incubation times were shortened at 30 min. Membranes were washed 3 times with TBS-T for 10 min and then treated with ECL reagent (Thermo Scientific). Development of blots was carried out with a Biospectrum Imaging System (UVP LLC, Upland, CA, USA) using the Vision Works LS software. Image analysis was carried out using the Quantity One (Bio-Rad, Hercules, CA, USA) software. Both irradiated and sham control data were expressed, normalizing the intensity of the protein of interest to the corresponding α-Tubulin or GAPDH band for each sample. Samples were analyzed in triplicates (Sup data WB blots).

### 5.8. Wound-Healing Assay (IncuCyte^®^ Live-Cell Analysis Systems)

In order to further study the impact of the conditioned medium on chondrocyte motility, a wound-healing test was performed on chondrocytes using the conditioned medium of chondrosarcoma cell X-rays irradiated with 0 Gy (as control) and 0.1 Gy (as treated sample). This conditioned medium was taken from chondrosarcoma cell after 6- and 24-h incubations. Then, chondrocytes motility was followed during 24 h in contact with these conditioned media. A wound-healing assay was applied to evaluate cell migration ability. Next, 1.5 × 104 T/C28a2 cells/well were seeded in 96-well IncuCyte^®^ ImageLock Plates in media. Cells were seeded at a density of 70 to 80%. After 24 h, cells were scratched by IncuCyte^®^ WoundMaker to build an artificial wound. Afterwards, the media was removed, cells were washed two times with PBS, and conditioned media were added on cells. Cells were cultured at 37 °C and 5% CO_2_ and monitored using an IncuCyte^®^ S3 (Sartorius). The migrating distance was measured for 24 h. Data were analyzed by the Cell Migration Analysis software module (Sartorius).

### 5.9. Statistical Analyses

Secretome statistical analysis. Spectral counts for each condition and each protein were normalized as recommended [39]. Abundances of the proteins, based on their normalized spectral counts, were compared according to the PatternLab Tfold comparison [40]. Only proteins with statistical significance (*p* value below 0.05) and with a Tfold increase or decrease in at least 50% compared to control were considered as differentially abundant. Two-dimensional-gels statistical analysis. Statistical analyzes were carried out following 3 independent experiments, using the *t*-test function of the Progenesis SameSpots software. Datasets were considered as significantly different when *p* < 0.05 (*). 8-oxo-dG statistical analysis. Statistical analyzes were carried out following 3 independent experiments, using the *t*-test function of the Excel Software in order to compare the 0 Gy condition to every other condition. Datasets were considered as significantly different when *p* < 0.05 (*). Western blotting statistical analysis. Statistical analyzes were carried out following 4 independent experiments, each made at least in 4 replicates, using the *t*-test function (= *t*-test) of the Excel Software in order to compare the 0 and 0.1 Gy condition, after normalization of the 0 Gy condition. Datasets were considered as significantly different when *p* < 0.05 (*). IncuCyte wound-healing migration test statistical analysis. Statistical analyzes were carried out following 2 independent experiments, each made at least in triplicats, using the *t*-test function (= *t*-test) of the Excel Software in order to compare each time point of 0 and 0.1 Gy condition. Datasets were considered as significantly different when *p* < 0.05 (*).

### 5.10. Mass Spectrometry Data

The mass spectrometry proteomics data were deposited to the ProteomeXchange Consortium via the PRIDE [41] partner repository with the dataset identifier PXD024953 and project doi:10.6019/PXD024953 in the case of secretome and PXD025187 in the case of proteome analyses.

In the case of secretome analysis, the reviewers may access this private dataset using reviewer_pxd024953@ebi.ac.uk as Username and iotcSWYp as Password.

In the case of proteome analysis, the reviewers may access this private dataset using reviewer_pxd025187@ebi.ac.uk as Username and imUEHlnL as Password. These data will be automatically accessible after publication.

## Figures and Tables

**Figure 1 ijms-22-07957-f001:**
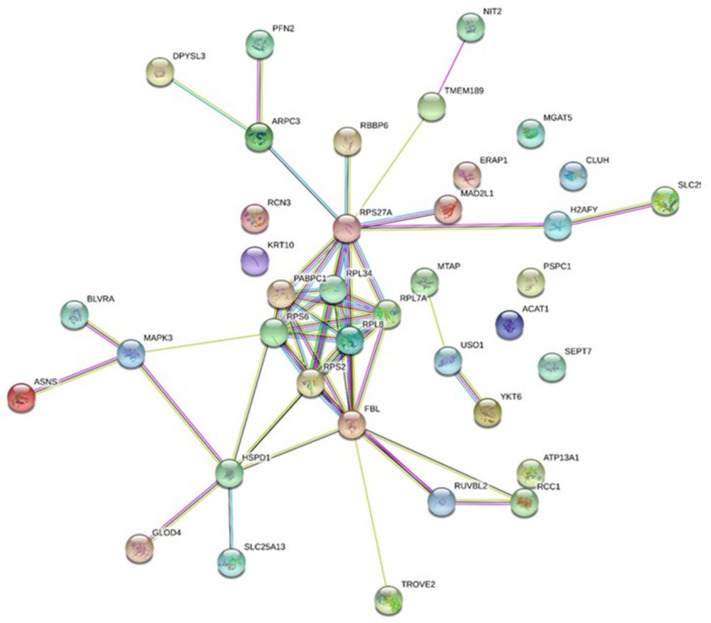
Bystander secretome network. Protein–protein interaction network constructed with protein accession from the list of 40 modulated proteins (24 up-regulated and 16 down-regulated) in the medium-conditioned/secretomic analysis. The network was constructed on STRING database.

**Figure 2 ijms-22-07957-f002:**
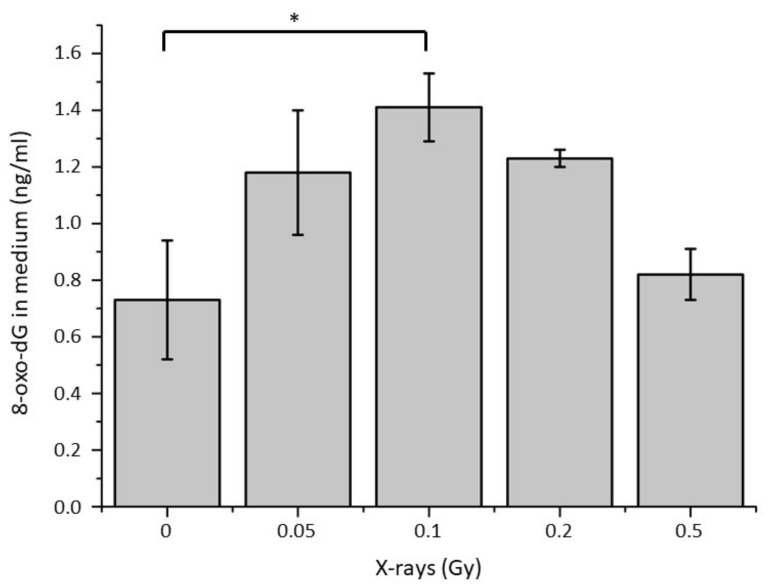
Quantification of 8-oxo-dG in the conditioned medium of chondrosarcoma cells irradiated with different doses of X-rays. (* = *p* < 0.05).

**Figure 3 ijms-22-07957-f003:**
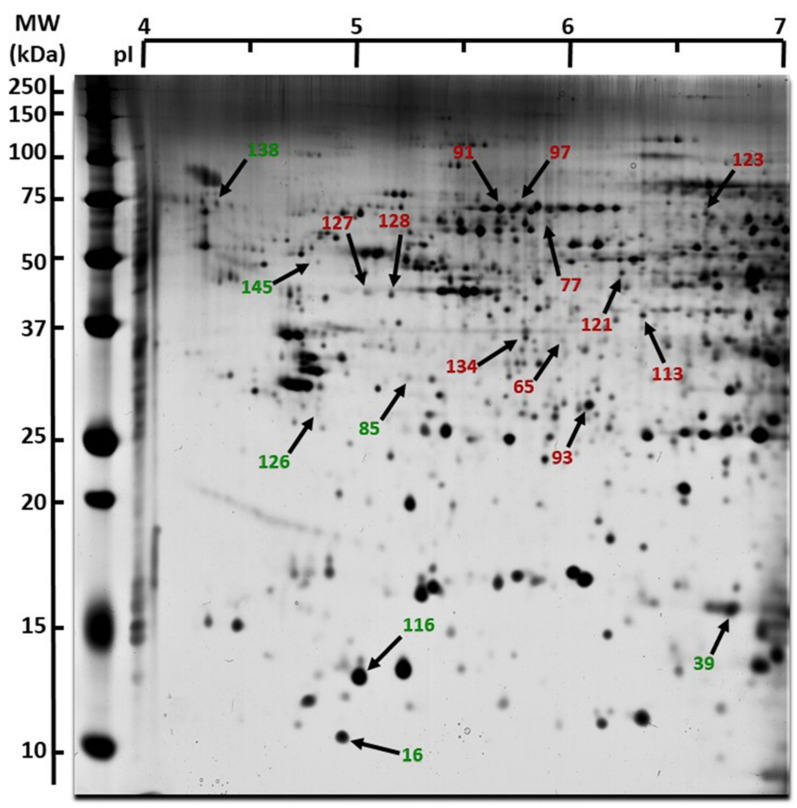
Proteome changes following bystander effect. Whole cell extracts from T/C-28A2 bystander cells receiving the conditioned medium of (1) low-dose irradiated chondrosarcoma cells or (2) non-irradiated chondrosarcoma cells were analyzed and compared by 2DE. One representative gel of (1) is shown. A total of 250 micrograms proteins were separated using 18-cm pH 4-7 pI range strips for the first dimension, and 12% acrylamide gels for the second dimension. Differentially expressed spots were delineated either in green (induced in (1) cells) or in red (repressed in (1) cells).

**Figure 4 ijms-22-07957-f004:**
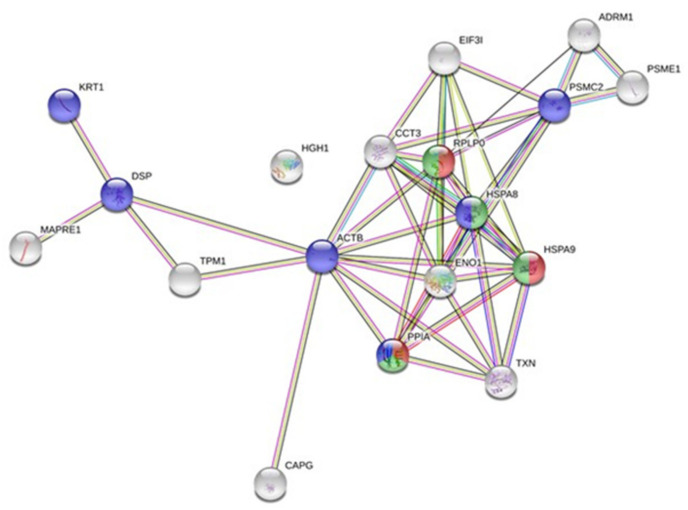
Bystander whole-cell proteome network. Protein–protein interaction network constructed with protein accession from the list of 20 modulated proteins (9 up-regulated and 11 down-regulated) in the bystander proteomic analysis. The network was constructed on STRING database.

**Figure 5 ijms-22-07957-f005:**
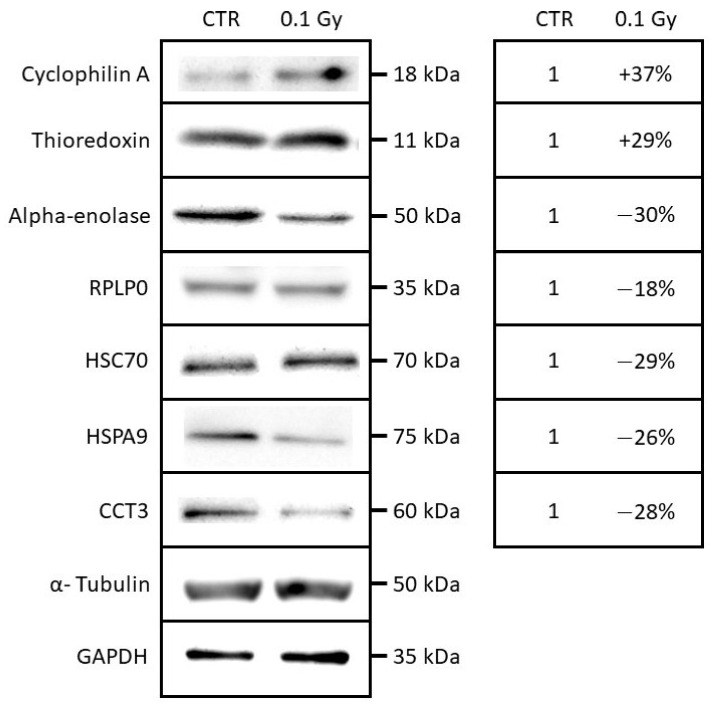
Western blotting analysis of cyclophilin A, thioredoxin, alpha-enolase, RPLP0, HSC70, HSPA9, and CCT3 in a whole-cell extracts from T/C-28A2 bystander cells receiving the conditioned medium of low-dose irradiated chondrosarcoma cells (0.1 Gy) or non-irradiated chondrosarcoma cells (CTR).

**Figure 6 ijms-22-07957-f006:**
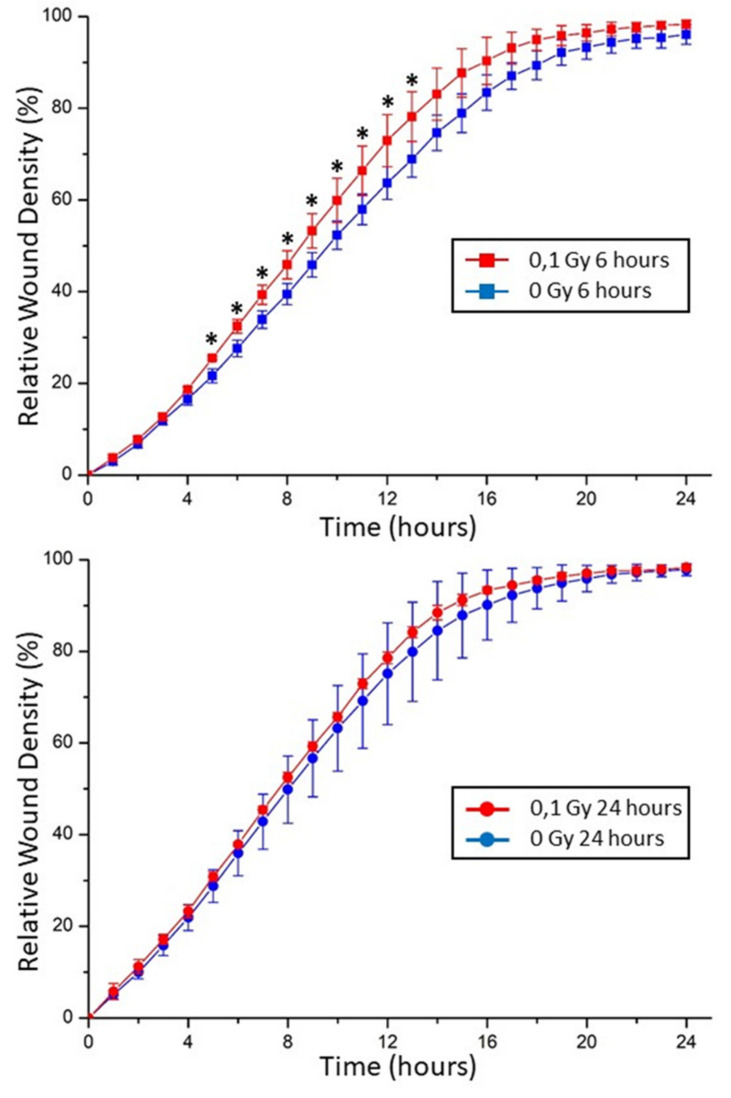
The conditioned medium of low-dose irradiated chondrosarcoma cells transiently increased the motility of T/C-28A2 bystander cells. Wound-healing assay showed the ability of cell migration in each group at 6 h (**top**) and 24 h (**bottom**) of contact with the conditioned media. (* = *p* < 0.05).

**Figure 7 ijms-22-07957-f007:**
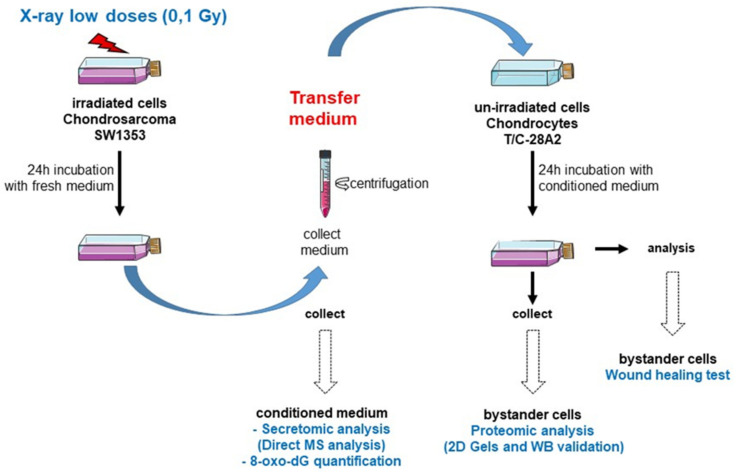
Schematic representation of experiments followed for the medium transfer protocols. Chondrosarcoma cells (SW1353 cell line) were irradiated at confluence in T25 flasks with X-rays or sham irradiated. Immediately after irradiation, the medium was changed with fresh new medium and incubated for 24 h. Then, the conditioned medium was centrifuged and collected. This conditioned medium (three independent biological replicates) was then analyzed for proteomic composition (secretomic analysis by direct MS analysis) or 8-oxo-dG quantification; or transferred to non-irradiated chondrocytes (T/C-28A2 cell line) for 24 h in T25 flasks with the same cell density. Then, cells were collected for proteomic composition (2D gel comparison and WB validation) or woundhealing test.

**Table 1 ijms-22-07957-t001:** List of 40 modulated proteins (24 up-regulated and 16 down-regulated) in the conditioned medium of low-dose irradiated chondrosarcoma cells, as compared with non-irradiated chondrosarcoma cells.

Accession	Name	pI	Mass (Da)	Tfold *	*p*-Value
P11940	Polyadenylate-binding protein 1	9.85	106.299	23.83	0.0264
P24752	Acetyl-CoAacetyltransferase, mitochondrial	8.39	58.871	4.00	0.0051
A5PLL7	Transmembraneprotein 189	9.07	107.703	3.33	0.0278
P08243	Asparagine synthetase [glutamine-hydrolyzing]	6.86	67.256	3.17	0.0155
P35080	Profilin-2	8.87	77.777	2.86	0.0423
P49207	60S ribosomal protein L34	10.64	21.824	2.50	0.0011
Q9HC38	Glyoxalase domain-containing protein 4	7.94	68.961	2.11	0.0191
P62424	60S ribosomal protein L7a	10.54	33.547	2.00	0.0150
P62753	40S ribosomal protein S6	10.74	31.799	2.00	0.0064
Q16181	Septin-7	8.97	244.012	2.00	0.0076
P27361	Mitogen-activated protein kinase 3	9.14	69.163	2.00	0.0250
O60763	General vesicular transport factor p115	5.88	159.184	1.91	0.0191
P62979	Ubiquitin-40S ribosomal protein S27a	9.64	19.523	1.84	0.0356
Q53SE2	Uncharacterized protein HSPD1	8.32	70.924	1.83	0.0186
P62917	60S ribosomal protein L8	11.15	32.789	1.79	0.0017
Q9NQR4	Omega-amidase NIT2	6.73	47.093	1.75	0.0335
P15880	40S ribosomal protein S2	10.37	34.399	1.67	0.0156
O75367	Core histone macro-H2A.1	9.75	68.531	1.67	0.0379
Q13126	S-methyl-5’-thioadenosine phosphorylase	9.17	186.699	1.64	0.0031
Q96D15	Reticulocalbin-3	5.05	46.220	1.64	0.0047
Q7Z6E9	E3 ubiquitin-protein ligase RBBP6	9.65	98.855	1.58	0.0385
O00370	LINE-1 retrotransposable element ORF2 protein	9.51	5.633.488	1.57	0.0179
Q9NZ08	Endoplasmic reticulum aminopeptidase 1	9.00	202.092	1.50	0.0219
P22087	rRNA 2’-O-methyltransferase fibrillarin	10.19	41.124	1.50	0.0219
O15145	Actin-related protein 2/3 complex subunit 3	9.60	28.754	−1.67	0.0379
Q13257	Mitotic spindle assembly checkpoint protein MAD2A	6.30	55.269	−1.67	0.0250
O15498	Synaptobrevin homolog YKT6	8.79	96.462	−1.90	0.0171
Q9UJS0	Calcium-binding mitochondrial carrier protein Aralar2	9.33	119.145	−2.00	0.0011
Q00325	Phosphate carrier protein, mitochondrial	9.34	63.151	−2.00	0.0409
Q8WXF1	Paraspeckle component 1	8.96	74.454	−2.08	0.0003
P10155	60 kDa SS-A/Ro ribonucleoprotein	9.63	350.653	−2.10	0.0160
Q5JXB2	Putative ubiquitin-conjugating enzyme E2 N-like	9.15	95.474	−2.11	0.0030
P53004	Biliverdin reductase A	6.47	41.158	−2.18	0.0405
Q9Y230	RuvB-like 2	5.40	54.097	−2.24	0.0169
O75153	Clustered mitochondria protein homolog	6.34	190.902	−2.29	0.0427
Q9HD20	Manganese-transporting ATPase 13A1	9.48	70.012	−2.50	0.0029
P18754	Regulator of chromosome condensation	5.79	42.947	−2.67	0.0108
Q09328	Alpha-1,6-mannosylglycoprotein 6-beta-N-acetylglucosaminyl transferase A	9.09	255.616	−2.75	0.0003
Q14195	Dihydropyrimidinase-related protein 3	8.69	171.819	−2.85	0.0245
P13645	Keratin, type I cytoskeletal 10	6.00	75.115	−2.87	0.0138

* Tfold is positive for accessions up regulated in the conditioned medium of irradiated cells, and negative for accessions down regulated in the conditioned medium of irradiated cells.

**Table 2 ijms-22-07957-t002:** List of 20 modulated proteins in the proteome of bystander cells receiving the conditioned medium of low-dose irradiated chondrosarcoma cells, as compared with bystander cells receiving the conditioned medium non-irradiated chondrosarcoma cells.

Spot Number	Fold	pI *	MW *	Accession	Names	HighestMean	GO—Biological Process
16	1.31	4.79	20	P60709	Actin	Bystander 0.1 Gy	Cell junction assembly
39	1.62	6.69	26	P62937	Cyclophilin A	Bystander 0.1 Gy	positive regulation of protein secretion; interleukin-12-mediated signaling pathway
85	1.38	5.11	63	Q15691	Microtubule-associated protein RP	Bystander 0.1 Gy	cell migration
93	1.31	5.99	57	Q06323	PSME1	Bystander 0.1 Gy	interleukin-1-mediated signaling pathway; tumor necrosis factor-mediated signaling pathway
116	1.27	4.87	23	P15924	Desmoplakin	Bystander 0.1 Gy	adherent junction organization; cell-cell adhesion
116				P10599	Thioredoxin		cell redox homeostasis;
126	1.63	4.67	56	P09493	Tropomyosin alpha-1 chain	Bystander 0.1 Gy	negative regulation of cell migration
138	1.98	4.18	145	P04264	Keratin, type II	Bystander 0.1 Gy	Extracellular exosome
145	1.47	4.67	98	Q9BTY7	Protein HGH1 homolog	Bystander 0.1 Gy	unknown (interact with Peptidyl-prolyl cis-trans isomerase and Heat shock protein 90)
65	1.81	5.86	72	P06733	Alpha-enolase	Bystander 0 Gy	negative regulation of cell growth
65				P05388	60S acidic ribosomal protein P0		interleukin-12-mediated signaling pathway
77	1.56	5.79	130	P01876	IGHA1 protein	Bystander 0 Gy	Extracellular exosome
91	1.29	5.63	140	P11142	HSC 70 protein	Bystander 0 Gy	cytokine-mediated signaling pathway
97	1.32	5.64	141	P38646	Hspa9	Bystander 0 Gy	interleukin-12-mediated signaling pathway
113	1.27	6.25	79	P40121	CAP-G protein	Bystander 0 Gy	Protein motility
121	1.47	6.15	94	P35998	26S proteasome regulatory subunit 7	Bystander 0 Gy	interleukin-1-mediated signaling pathway
123	1.31	6.56	136	P49368	T-complex protein 1 subunit gamma	Bystander 0 Gy	Extracellular exosome
127	1.31	4.91	87	P60709	Actin	Bystander 0 Gy	Cell junction assembly
128	1.36	5.03	87	Q16186	Proteasomal ubiquitin receptor ADRM1	Bystander 0 Gy	Proteasome complex
134	1.26	5.68	73	Q13347	Eukaryotic translation initiation factor 3 subunit I	Bystander 0 Gy	Extracellular exosome

*: pI (iso-electric point) and MW (kDa) according to the 2D gel location. There were 9 up-regulated (highest mean with Bystander 0.1 Gy) and 11 down-regulated (highest mean with Bystander 0 Gy).

## Data Availability

Not applicable.

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
