# Peer review of "A Proteomic Study Suggests Stress Granules as New Potential Actors in Radiation-Induced Bystander Effects"

_ijms, 2021, doi:10.3390/ijms22157957_

Round 1

Reviewer 1 Report

This is an interesting manuscript which describes a proteomic study of radiation induced bystander factors. Analysis of bystander cells and of conditioned media has been performed and a number of proteins were identified and validated. In addition, some functional assays were performed.

Interestingly, proteins involved in stress granules were identified in the conditioned media but not in the bystander cells. Please discuss this and suggest why this could be.

Please include other relevant literature on proteomic analysis of bystander cells and conditioned media in the Introduction and Discussion sections.

Please describe the irradiation and dosimetry in more detail.

Reviewer 2 Report

Manuscript (MS)    IJMS – 1291943

Title: “A proteomic study suggests stress granules as new potential actors in radiation-induced bystander effects.”

Authors: Mihaela Tudor, Antoine Gilbert, Charlotte Lepleux, Mihaela Temelie, Sonia Hem, Jean Armengaud, Emilie Brotin, Siamak Haghdoost, Diana Savu and François Chevalier (Corresponding).

Overall comments:

The manuscript (MS) represents an original research report. The objectives of the conducted research were to investigate the mechanisms the intercellular communication between the irradiated chondrosarcoma cells and the non-irradiated chondrocytes. For this purpose the Authors deployed the advanced proteome techniques to analyze the secretome (i.e., “the bystander factors”) of irradiated chondrosarcoma cells and the proteome of the chondrocytes exposed to the above “bystander factors”. As stated the implemented approaches allowed the Authors [q] “to propose new bystander candidates and cellular responses, potentially involved in these non-targeted effects”.  Overall, the presented research is well-designed and well-developed. Manuscript is reasonably well-written.

Based on the advanced authentication analysis (iThenticate) there is minimum (8%) cross-match with the Author’s previously published papers that occurs in Introduction and Materials and Methods of current MS. However, there are some key figures/images presented in this communication which have been recently reported on-line (HAL Archives, https://hal.archives-ouvertes.fr/hal-03144446), namely Table 1 (Secretomic analysis of conditioned medium from chondrosarcoma cells), Fig. 3 (Proteomic analysis of bystander cells), Fig. 6 (a graph representing a secretome network), Fig 7 (a graph representing a proteome network in chondrocytes exposed to the “bystander factors”).

Then, since the senior Author of this MS is the Guest Editor of the special issue that would be at his discretion how to manage the above comments.

Specific comments

LL 25-28

“Besides the direct effects of radiations, indirect effects are observed within the surrounding non-irradiated area; irradiated cells relay stress signals in this close proximity inducing the so-called radiation-induced bystander effect. These signals received by neighboring unirradiated cells induce specific responses similar with those of direct irradiated cells.”

And:

LL 106-108

“In order to study the bystander effect between irradiated chondrosarcoma cells (SW1353 cell line) and non-irradiated chondrocytes (T/C28-A2 cell line), we selected a medium- transfer protocol, and we kept the same treatment strategy with all our endpoints (Fig. 1).”

While implementation of the described “medium- transfer protocol” was justifiable for the purpose of the reported research, technically, in the conducted experiments the chondrocyte cells were not “bystander” toward irradiated chondrosarcoma cells.

What I am implying is - with the same success the authors could define the implemented cell model as “investigation of abscopal effect”. Note that the cartilaginous tissue is present in various parts of the skeleton and extracellular vesicles can exert endocrine-like effects.  In this respect, perhaps, the Author would re-consider using “NTE” instead of “bystander” effects?

Minor Comments:

Table 1 and Table 2 – define units for “separation values” clarify where it was observed up-regulation and down regulation.

Under Materials and Methods;

Indicate sources and nature of all primary antibodies.

There are numerous mistypes; try some professional help to clear MS.

e.g., “the bystander effectinduced by low doses irradiated”

Reviewer 3 Report

Radiation-induced bystander effects are very important biologic responses after irradiation, however the underlying mechanism has not yet been identified. This manuscript provides valuable information about the bio-physiologic mechanism of radiation-induced bystander effects. The overall quality of this manuscript is good and methodology is also appropriate. I recommend some points should be revised for making better manuscript. 

1) Overall, the word spacing is not correct. Authors should check it carefully.

2) 2 pages 90-92 lines. "The bystander factors.... a gel-based strategy"

This paragraph should be moved to Methods section

3) 3 pages 106-117 lines

In Results section, only study results should be described. This paragraph should be moved to Methods section

4) 4 pages 132-141 lines

This paragraph should be moved to Methods section

5) 10 pages 269-274 lines

This paragraph should be moved to Methods section

6) 11 pages 296-306 lines

In Discussion section, you have to describe the discussion on the results you described in Results section. This paragraph is should be moved to Results section.

7) 13 pages 383-388 lines

This paragraph is should be moved to Results section.

Round 2

Reviewer 1 Report

Thanks to the authors for addressing all the comments satisfactorily.